# Reliable long-term individual variation in wild chimpanzee technological efficiency

S. Berdugo [1,2,3] ✉, E. Cohen[1,2], A. J. Davis[1,2,4], T. Matsuzawa [5,6] & S. Carvalho[1,3,7,8]

Variation in the efficiency of extracting calorie-rich and nutrient-dense resources directly impacts energy expenditure and potentially has important repercussions for cultural transmission where social learning strategies are used. Assessing variation in efficiency is key to understanding the evolution of complex behavioural traits in primates. Here we examine evidence for individual-level differences beyond age- and sex-class in non-human primate extractive foraging efficiency. We used 25 years (1992–2017) of video of 21 chimpanzees aged ≥6 years in Bossou, Guinea, to longitudinally investigate individual-level differences in stone tool use efficiency. Data from 3,882 oil-palm nut-cracking bouts from >800 h of observation were collected. We found reliability in relative efficiency across four measures of nut-cracking efficiency, as well as a significant effect of age. Our findings highlight the importance of longitudinal data from long-term field sites when investigating underlying cognitive and behavioural diversity across individual lifespans and between populations.

The importance of individual variation in cognition and behaviour is increasingly appreciated in research on non-human animals[1]. Such variation can have ramifications at both the individual and population levels, with broader implications for life history[2–4], cultural evolution and interpretations of the archaeological record[5] (Fig. 1). For example, a recent large-scale meta-analysis found individual variation in the migration timing of land-, water- and seabirds, which impacts the breeding success and survival of a given migration[6].

Yet, although cognitive and behavioural variation has been reported, the research has mostly focused on one time point for each individual or is limited to short time spans[1,7,8]. We argue that this approach does not reflect the true extent and patterns of variation within wild populations. Here we aim to fill this knowledge gap, investigating individual differences in chimpanzee technological efficiency and their persistence over time.

First, persistent variation in the proficiency and efficiency with which individuals extract high-quality resources from the environment directly impacts energy expenditure and individual fitness. An individual who is slow or inefficient at a given extractive-foraging technique relative to others in the population will have less time and energy for other fitness-enhancing tasks, thus incurring relative fitness costs[9]. This is particularly true for complex tool-assisted foraging tasks aimed at extracting high-calorie resources, such as wood-boring beetle larvae extraction by New Caledonian crows (*Corvus moneduloides*)[10], oyster-cracking by Burmese long-tailed macaques (*Macaca fascicularis aurea*)[7], and honey-dipping, termite-fishing and nut-cracking by chimpanzees (*Pan troglodytes* spp.)[8,11,12]. Understanding the extent and causes of individual differences in extractive-foraging tasks is key for identifying variation in factors known to be relevant to evolutionary fitness.

Second, there is growing recognition that variation in tool use produces variation in the traces left in the archaeological record, particularly for the signatures produced during percussive behaviours[5].

[1]School of Anthropology and Museum Ethnography, University of Oxford, Oxford, UK. [2]Centre for the Study of Social Cohesion, School of Anthropology and Museum Ethnography, University of Oxford, Oxford, UK. [3]ICArEHB, Interdisciplinary Center for Archaeology and Evolution of Human Behaviour FCHS, Universidade do Algarve, Faro, Portugal. [4]Wadham College, University of Oxford, Oxford, UK. [5]Department of Pedagogy, Chubu Gakuin University, Gifu, Japan. [6]College of Life Science, Northwest University, Xi'an, China. [7]Department of Science, Gorongosa National Park, Sofala, Mozambique. [8]CIBIO/InBIO, Centro de Investigação em Biodiversidade e Recursos Genéticos, Universidade do Porto, Campus de Vairão, Vairão, Portugal. ✉e-mail: sophie.berdugo@anthro.ox.ac.uk

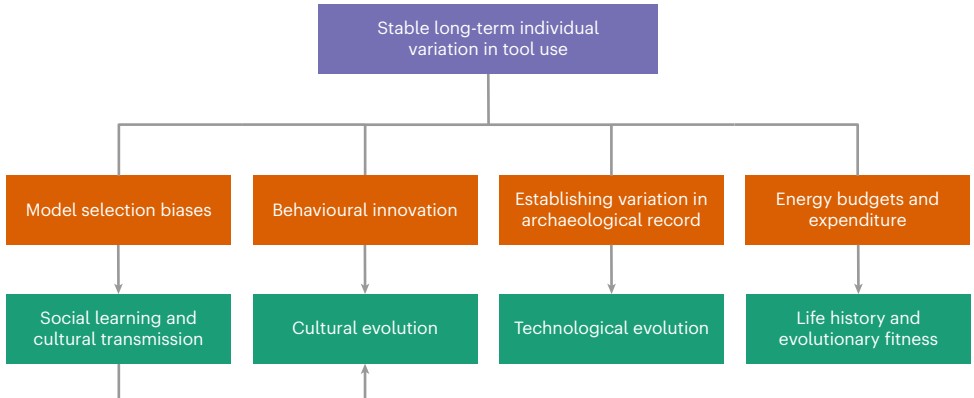

**Fig. 1 | Implications of long-term stable individual variation in lithic technological behaviour.** The schematic illustrates the implications of long-term individual differences in stone tool use on four major domains (orange): the potential presence of social learning strategies, the generation of behavioural innovations, the interpretation of the archaeological record, and the presence of variation in daily energy budgets and expenditure. In turn these have four broader evolutionary implications (green): transmission via social learning, cultural evolution, technological evolution, and life history strategies and evolutionary fitness. Arrows indicate the relationships between the major domains and the broader evolutionary implications.

For example, the number of unintentional flakes produced by wild bearded capuchin monkeys (*Sapajus libidinosus*) while cracking nuts with stones will vary according to the tool user, with the skill (or lack thereof) of the individual determining the frequency of mishits and subsequent flakes[13]. Investigating the archaeological signatures of known individuals can inform our understanding of hominin fossil sites and the context behind technological traces[5]. However, there is currently no research on individual-level variation in chimpanzee stone tool use, constituting a substantial gap in our understanding of primate technological behaviour and its potential implications for hominin tool use.

Third, identifying systematic variation in technological behaviour is crucial for investigating the factors influencing individual differences in behavioural acquisition via social learning. Indeed, although there are species-typical forms of social learning for particular skills, there is growing evidence that social learning mechanisms vary systematically across individuals[14]. Individual-level variation can develop from 'individual learning of social learning', whereby differing experiences with previous social learning opportunities can result in different strategies being used in the future, or the rate of learning being altered. This phenotypic plasticity can facilitate faster adaptation to environmental changes, potentially having profound effects on evolutionary processes. This is particularly true where model selection and social learning biases guide behavioural acquisition[15–18]. For example, female migrant chimpanzees in the Taï Forest, Côte d'Ivoire, conform to the nut-cracking technique of their new community, even when their previous technique was more efficient (in terms of strikes per nut and foraging speed)[17]. Therefore, assessing variation contributes to our understanding of cultural transmission and the evolution of complex behavioural traits in primates[19].

Variation in technological behaviour in non-human primates is well established. For example, only male bearded capuchins use sticks as probing tools[20]. In chimpanzees, inter- and intra-population variation in tool use is found across all communities, including differences in technological strategies and efficiency. For instance, ant-dipping tool lengths differ between the two neighbouring communities in the Kalinzu Forest, Uganda[21]. Chimpanzees in Kibale National Park, Uganda, used sticks to extract experimentally introduced honey, whereas chimpanzees in Budongo National Park, Uganda, either used their fingers or leaf sponges[22]. Chimpanzees in Loango National Park, Gabon, use different grip types to perforate nests during honey extraction, which is unaffected by the soil hardness[8]. In Gombe National Park, Tanzania, females commence termite-dipping earlier, engage in fishing more frequently and retrieve more termites per dip than males[23].

**Table 1 | Definitions of efficiency measures**

| Efficiency measure | Definition |
| --- | --- |
| Bout duration | The length of the continuous period (in seconds) of nut-cracking whereby the individual strikes a single nut on an anvil with a hammer stone involving the same hand, bodily posture, hand grip and nut (the 'bout')[31]. A bout starts in the frame when the hammer is lifted for the first strike of the nut on the anvil and ends in the frame when the hammer makes contact with the nut for the final time[50]. |
| Strikes per nut | The number of hits required to open a single nut using a stone hammer and stone anvil[24]. |
| Success rate | The outcome of the nut-cracking bout. A successful bout is one where the nut is cracked using the stone hammer-and-anvil composite and the full kernel is retrieved and eaten. A failed bout is one where the nut is not retrieved or eaten by any individual. A smashed nut is one where the nut is opened but the kernel has broken into multiple pieces such that each piece must be eaten separately[38]. |
| Displacement rate | The number of times a hammer strike resulted in the nut being displaced from the anvil[19]. |
| Tool switch rate | The number of times the focal individual altered their tools while cracking nuts[19]. |

In nut-cracking, females in Taï crack *Coula edulis* and *Panda oleosa* nuts more efficiently (measured as the mean number of strikes per nut and nuts per minute) than males[24,25]. By contrast, males in Bossou, Guinea, select, use and transport tools more frequently than females[26], and spend a significantly greater proportion of their time cracking oil palm (*Elaeis guineensis*) nuts compared with adult females[27]. The number of hits required to successfully open the nutshell was previously found to reach an asymptote in adulthood[28], and the movement of adult crackers was considered to be stereotyped[29]. However, this finding is constrained to a measure of strikes per nut, which is only one measure of nut-cracking efficiency. Moreover, no research has sought to replicate this finding, and the presence and stability of variation in nut-cracking behaviour has not been investigated using long-term data.

The combination of chimpanzees' long lifespans and the protracted learning periods (3–5 years for nut-cracking in Bossou[29,30]) required to learn complex tool use necessitates longitudinal studies for assessing developmental trajectories and social learning. Although cross-sectional data can be useful for determining general developmental milestones, they do not allow for tests of individual variation across the lifespan. By obscuring potential variability, this may result

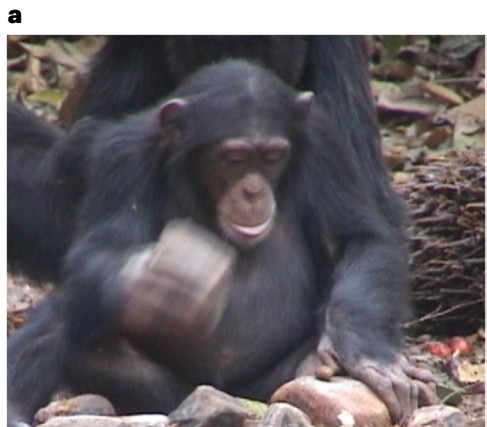
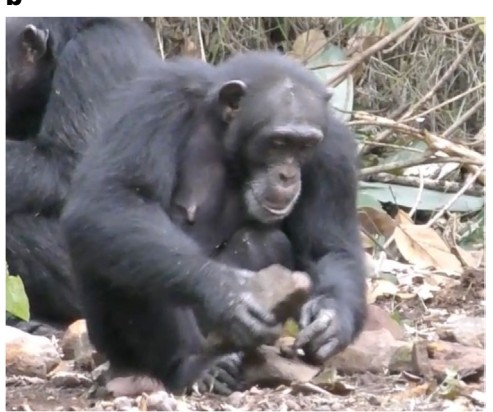

**Fig. 2 | Stills taken from the Bossou video archive. a**, A 6-year-old female chimpanzee, Fanle, cracks an oil palm nut in 2004, the year after finishing her early learning period. **b**, The same female chimpanzee cracks an oil palm nut in 2017 at the age of 20. © Sophie Berdugo and Tetsuro Matsuzawa.

in generalizations about population- or species-typical development of a behavioural trait. Tracking the development of a skill across an individual's life course is key for understanding the ontogenetic processes involved–and potential variation in–fitness-enhancing traits, including extractive technologies.

This study assessed five measures (Table 1) of post-early learning period (hereafter, 'post-ELP') nut-cracking efficiency for 21 individuals present in a 25-year video archive of wild chimpanzees. 'Post-ELP' was defined as chimpanzees aged 6 and over, based on the previously established period of social learning lasting until 5 years old[29]. Data were gathered from all post-ELP individuals present in the archive for each year they were in the footage (Fig. 2 and Supplementary Table 1). This research takes a longitudinal approach to investigate the presence of variation in primate extractive foraging. We also assess the reliability (internal consistency across efficiency measures) and stability (whether individual differences hold over time)[1] of such differences in a wild primate population.

## Results

### Nut-cracking bouts
Bouts are the continuous periods (in seconds) of nut-cracking whereby the individual strikes a single nut on an anvil with a hammer stone involving the same hand, bodily posture, hand grip and nut[31]. A total of 3,882 oil-palm nut-cracking bouts were recorded across 21 chimpanzees (12 females and 9 males; ages 6–60) over a 25-year period. Of these bouts, 318 (8.17%) ended in failure (no kernel extracted), 336 (8.66%) ended with a smashed kernel (only partial retrieval of the kernel) and 3,228 (83.17%) ended in the successful retrieval of a whole kernel.

### Individual variation
Analyses (see Supplementary Tables 2–6 for full model outputs) revealed that including random intercepts for individual chimpanzees improved model fit for all measures: log bout duration ($\chi^2(1) = 369.49$, $P < 0.0001$), strikes per nut ($\chi^2(1) = 475.26$, $P < 0.0001$), success rate ($\chi^2(1) = 47.16$, $P < 0.0001$), displacement rate ($\chi^2(1) = 87.022$, $P < 0.0001$) and tool switch rate ($\chi^2(1) = 4.17$, $P = 0.0411$). Age had a significant positive fixed effect on log bout duration ($t(1400) = 7.724$, $P < 0.001$, $\beta = 0.02$, 95% confidence interval (CI) = (0.016, 0.028)) and strikes per nut ($z(3362) = 8.51$, $P < 0.001$, $\beta = 0.024$, 95% CI = (0.019, 0.030)), and a significant negative fixed effect on displacement rate ($z(3666) = -2.40$, $P = 0.0164$, $\beta = -0.012$, 95% CI = (-0.022, -0.002)) and tool switch rate ($z(3666) = -2.462$, $P = 0.0138$, $\beta = -0.010$, 95% CI = (-0.018, -0.002)). There was a significant fixed effect of sex for tool switch rate (male, $z(3666) = -2.502$, $P = 0.0124$, $\beta = -0.384$, 95% CI = (-0.685, -0.083)), but no significant fixed effect of sex for log bout duration (male,

$t(16) = 0.238$, $P = 0.812$, $\beta = 0.07$, 95% CI = (-0.520, 0.670)), strikes per nut (male, $z(3362) = 0.407$, $P = 0.684$, $\beta = 0.114$, 95% CI = (-0.436, 0.664)), success rate (male, $z(3668) = -0.1296$, $P = 0.897$, $\beta = -0.025$, 95% CI = (-0.406, 0.356)) or displacement rate (male, $z(3666) = -1.029$, $P = 0.304$, $\beta = -0.211$, 95% CI = (-0.612, 0.190)). Data distributions for each individual across the five efficiency measures can be found in Supplementary Figs. 1–5.

### Reliability of individual variation
Next, we ranked the random intercepts for each multilevel model; random intercepts represent estimated individual-level effects on the outcome variable in the model. Lower ranks represent greater relative nut-cracking efficiency: fewer strikes per nut, shorter bout durations and so on. Ranks ranged from 1 to 21 (the total number of individuals in the dataset; Fig. 3). Individual ranks for log bout duration, strikes per nut, success rate and displacement rate were strongly correlated (mean correlation, $r = 0.718$; Supplementary Fig. 6), meaning that when an individual was ranked highly on one of these measures, they also ranked highly in the other three. However, tool switch rate was only moderately correlated (mean correlation, $r = 0.319$), suggesting that it reflects a different underlying construct.

The consistency of the relative rank of each individual's random intercepts obtained in the log bout duration, strikes per nut, success rate and displacement rate models indicates reliable individual differences in nut-cracking efficiency. A two-way intra-class correlation (ICC) analysis also indicated good agreement, $F(20,60) = 11.2$, $P < 0.001$, ICC(A,1) = 0.728, 0.556 < ICC < 0.863. Including tool switch rate in the two-way ICC decreased the internal consistency of the efficiency measures ($F(20,80) = 7.32$, $P < 0.001$, ICC(A,1) = 0.57, 0.377 < ICC < 0.76), further suggesting that it reflects a different underlying construct.

### Stability of inter-individual relative efficiency
We also assessed whether individuals' relative nut-cracking efficiency was stable over time. To do this, we plotted the data and random slopes for the effect of age on the outcome of interest for individuals who had at least 3 years of data during what we call the 'adult proficiency period' (hereafter, APP): the period from ages 11 to 40. The APP encompasses the period of adulthood from maturity to the start of declining efficiency owing to old age (40 years old; Howard-Spink, E., Matsuzawa, T., Carvalho, S., Hobaiter, C., Almeida-Warren, K. et al., unpublished manuscript)[32] and also reflects the continued intra-individual improvement in efficiency from ages 6 to 10 before maturity is reached (Supplementary Figs. 1–5). By looking at individuals with at least 3 years of data, we were able to determine whether the random slope lines intersected one another or not, independent of the effects of age on efficiency

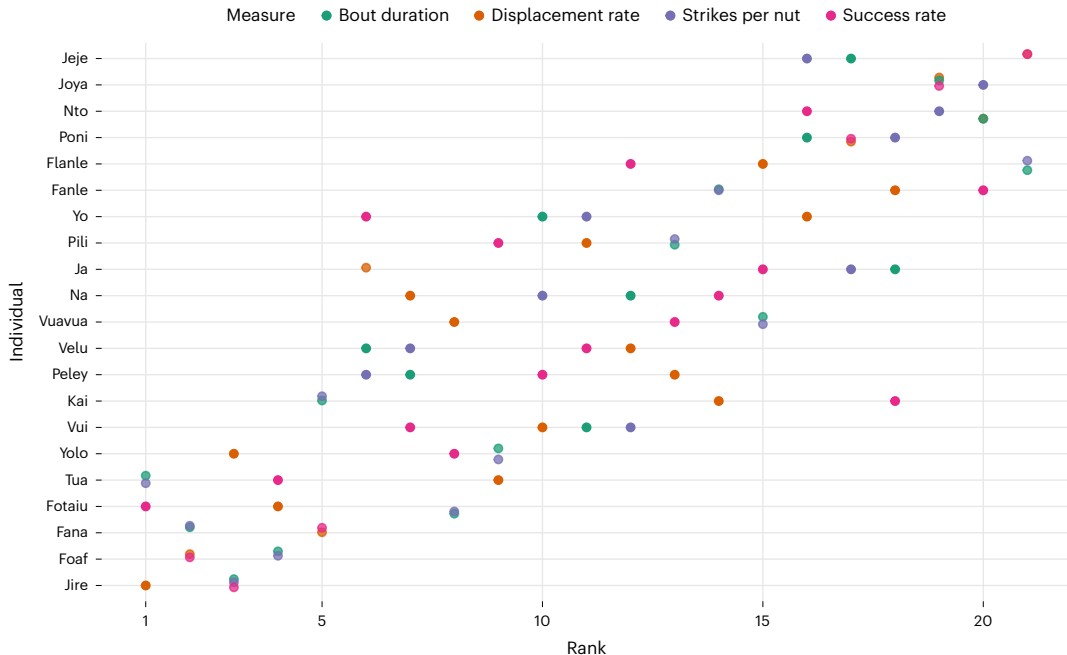

**Fig. 3 | Ranked random intercepts from each multilevel model for the four reliable measures of nut-cracking efficiency.** The relative nut-cracking efficiency ranking for all chimpanzees (n = 21). Lower values indicate greater relative nut-cracking efficiency (the chimpanzee with a rank of one is more efficient than the chimpanzee with a rank of two for that efficiency measure and so on).

reported above. In other words, we asked: even if individual X's efficiency improved or worsened over time, were they always more or less efficient than individual Y?

This dataset (n = 7; 3 males, 4 females; between 1,047 and 1,123 bouts in total, depending on the outcome measure) spans the APP, with a younger cohort of two individuals (1 male, 1 female, aged 11–20), one individual across the middle period (male, aged 12–37) and an older cohort of four individuals (1 male, 3 females, aged 34–40). The individuals in the younger cohort were both born in 1997 and their APP data spans 2008–2017, and the older cohort were all a similar age at the start of the study period (32–36 years old in 1992) and had APP data from 1992 to 1999. The middle individual was born in 1980 and did not overlap in age with any other individual during the same years in our dataset. The presence of multiple individuals of the same or similar ages overlapping in time allows for direct comparisons of relative efficiency within these two cohorts.

We were able to create models with random slopes for age for strikes per nut and displacement rate (models for log bout duration and success rate did not converge or had issues with singularity, respectively). The results are shown in Fig. 4. Although more data are needed for inferential statistics, we note that estimated random slopes for individuals within age cohorts do not intersect, suggesting little change in relative efficiency over time within these groups.

### Inter-rater reliability

Two independent, hypothesis-blind coders reviewed 70 h (8.41%) of footage for inter-coder reliability analyses. Unweighted Cohen's κ and ICC analyses indicated substantial–excellent agreement and consistency between coders (Supplementary Table 7).

### Discussion

This longitudinal research establishes reliable individual differences in efficiency across four measures of nut-cracking technological efficiency in wild chimpanzees. This finding highlights the necessity to move beyond the exclusive use of group averages when investigating daily energy expenditure and activity budgets–factors with important implications for life history and foraging ecology in primates. Variation

in the time and energy that an individual expends on nut-cracking could produce variation in resource allocation to other fitness-enhancing traits. Thus, the extent of variation in nut-cracking efficiency may have large implications for other factors impacting survival and reproduction at the individual level, such as social learning, as well as the evolution of cultural traits.

Moreover, this research demonstrates individual-level variation in chimpanzee stone tool use. This finding reiterates the need to consider variation in the archaeological signatures left by percussive technological behaviours, with some individuals potentially contributing more to the record than others[5,33]. Research is now needed to ascertain the sources of this variation. A particular focus on the nut-cracking learning period is key given the known developmental drivers of variation in technological behaviour[23,34,35].

The ranked random intercepts for log bout duration, strikes per nut, success rate and displacement rate were closely correlated, but the raw scores were not perfectly correlated. For example, the correlation between the ranked random intercepts for log bout duration and strikes per nut was greater than the correlation between their raw scores (r = 0.99 and r = 0.756, respectively). Indeed, longer bouts do not necessarily equate to more strikes per nut, with bout duration also being extended owing to factors such as greater distractibility or taking longer pauses between strikes. This supports the view that each measure captures distinct, but internally consistent, aspects of what can be termed 'efficiency'. Bout duration is an indicator of energy intake, as shorter bouts allow more kernels to be retrieved over the course of a feeding session. This may improve an individual's food security, as they will be at a competitive advantage compared with conspecifics in terms of access to more nuts. Also, longer bouts mean that the individual's attention is focused on the task for a relatively longer period of time, such that there is reduced time to be allocated to other fitness-enhancing behaviours such as defence, socialization and grooming.

Conversely, the number of strikes required to retrieve the kernel and the success rate of the individual are indicators of energy expenditure relative to energy intake. The number of strikes per nut also dictates how convenient it is for the individual to use stones to

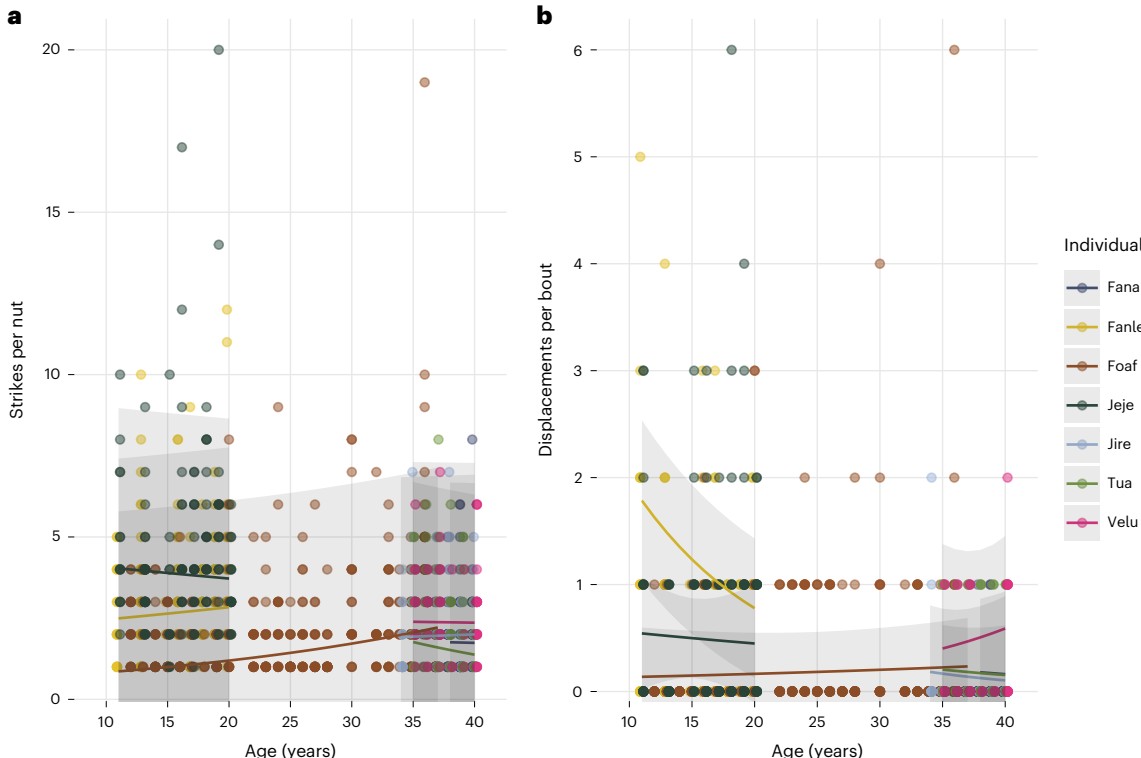

**Fig. 4 | Predicted random slopes for age for individuals within the adult proficiency period. a**, The number of strikes per nut for each individual over time. **b**, The number of displacements per nut for each individual over time. Lines represent predicted values and error bands represent ±s.e.

open the nuts (rather than, for example, using their teeth or scrounging kernels), with more convenient techniques being favoured owing to their increased efficiency[36]. For success rate, future research should investigate why chimpanzees choose to end a nut-cracking bout before the kernel is extracted and assess whether certain individuals stop bouts earlier than others (that is, do not expend unnecessary energy and are hence more efficient).

Displacement rate measures the ability, or lack thereof, to judge the amount of kinetic energy required to strike a nut. Displacing the nut not only expends more energy than is required, but also elongates the bout duration, particularly if the individual has to travel to retrieve the nut. This also reduces the convenience of using tools. Primate archaeological research should investigate whether individuals with higher displacement rates are also more likely to fracture their stone tools, resulting in potential unintentional flakes[13,37].

Our results also suggest that the number of times individuals adjust their tools during a bout reflects a different underlying process. Tool switch rate is a proxy measure for the individual's ability to select an efficient tool composite and position the stones in such a way to reduce the energetic effort required to extract the kernel. Given the importance of tool properties in determining the efficiency of the behaviour[17,38,39], learning to select the best tools is also an important aspect of successful foraging, although our results suggest that it does not relate to how efficient the individual is at cracking nuts per se.

Moreover, our results suggest that relative inter-individual nut-cracking efficiency (in terms of the number of strikes per nut and the displacement rate) for a subset of seven chimpanzees may be stable over time during the APP, although we call for more research on this topic using more robust datasets. Despite our data being relatively sparse—our models included only seven individuals—we were able to estimate the trajectory of individuals' nut-cracking efficiency over time within age cohorts who overlapped in the years they were present in the Bossou archive. Model estimates suggest that an individual's efficiency relative to others in its cohort persisted across overlapping

ages; individuals' predicted random slopes for the effects of age on the efficiency outcomes did not intersect. These model predictions are thus consistent with the assertion that individuals' relative efficiency ranks remain stable over time. However, given the relatively small sample of individuals, these findings should be viewed as suggestive and an avenue for future research.

The presence of stable variation in the different measures of nut-cracking efficiency could indicate variation in the underlying cognitive and/or motor capacities. For example, longer bout durations may signify that certain individuals are prone to distraction rather than an individual being inherently less skilled. Therefore, the potential long-term stability in individual variation in nut-cracking efficiency may suggest that there are stable cognitive differences between the individuals, which potentially has implications for performance in other key fitness-enhancing behaviours. Future research should assess whether variation exists in other behaviours in the community to determine the extent of these cognitive differences and their contributions to fitness-relevant behaviours and outcomes.

This research did not seek to directly test whether increased nut-cracking efficiency corresponded to relative fitness gains as the presence of confounds hinders our ability to establish a causal link between tool use efficiency and longevity or reproductive success. For example, a shorter lifespan of a relatively less efficient individual could be the result of technological inefficiency or another factor that negatively impacts both tool use efficiency and longevity (for example, physical weakness or ill-health). Research specifically investigating the fitness consequences of individual differences in technological efficiency should consider the broader energy budgets and expenditure, as relatively inefficient nutcrackers may make up for their energy disadvantages through increased efficiency in other tasks.

Age had a significant fixed effect on log bout duration, strikes per nut, displacement rate and tool switch rate. Increasing age corresponds to longer bouts and more strikes per nut, which may reflect greater muscle weakness and the need for more rests in old age. Indeed, there

have been many elderly chimpanzees in the Bossou population, with six individuals in the archive being aged 45+. Conversely, increasing age also corresponds to fewer nut displacements and fewer tool switches during bouts. This may reflect older individuals' greater proficiency at selecting appropriate tools, and better ability at judging the amount of kinetic energy required for each strike. More research is needed to assess the impact of old age on chimpanzee technological capabilities.

Consistent with previous research demonstrating that an individual's performance continues to improve until around age 10 (ref. 40), we show that an individual's efficiency increases until they reach maturity at age 11. Performance often improves and becomes less variable at ages 11–20 compared with at ages 6–10, suggesting that learning extends beyond the originally conceived learning period lasting until around 5 years old[29,41]. Although the initial acquisition of the skill relies on social learning from knowledgeable group mates[29], individuals' proficiency is honed through practice during the sub-adult period.

Unlike previous research, we found inconsistent effects of sex on technological efficiency. Sex had no significant effect on log bout duration, strikes per nut, success rate or displacement rate, but there was a significant effect on tool switch rate, with male chimpanzees switching tools less frequently than female chimpanzees. This finding contradicts previous research establishing a female bias in technological behaviour across the genus *Pan*[42]. This may reflect circumstances specific to the Bossou chimpanzees. Indeed, Bossou is an isolated community that has been decreasing in size, with both factors potentially contributing to unusual patterns compared with other populations. The lack of sex differences in this research may suggest that the mechanism(s) establishing a female bias in nut-cracking efficiency in Taï[24] are not present in Bossou. However, it may be that the apparent female bias is an artefact of short-term, cross-sectional analyses and that the effect does not hold longitudinally[43]. This reiterates the importance of conducting long-term investigations of long-lived primates as data from short-term studies may not be representative of behaviour over the lifespan. Further long-term research is required to determine whether this finding holds for other populations of nut-cracking chimpanzees.

The research is limited by three main constraints. First, the outdoor laboratory is a field experimental set-up, with locally sourced nuts and stones being provisioned, and the experiments only occurring during the dry season (when there is high fruit availability). It could be argued that this decreases the validity of the findings and that they may not reflect the year-round nut-cracking efficiency of the individuals (such as in periods of low fruit availability). However, the nut-cracking that occurs in the outdoor laboratory is no different from the nut-cracking that the Bossou chimpanzees perform at the natural cracking site of Moblim (07° 38′ 20.7″ N, 008° 30′ 39.2″ W)[26]. Moreover, the location of the outdoor laboratory was specifically selected to be on Mount Gban's summit—the core of the home range for this chimpanzee community—to optimize the frequency of chimpanzees visiting the site[40,44]. This suggests that nut-cracking behaviour remains unchanged regardless of site location, reiterating the ecological validity of the outdoor laboratory.

Second, despite the experimental nature of the outdoor laboratory, there are risks of confounding variables. For example, there is no control for daily intake of energy from other sources or levels of physical activity, both of which could influence motivations for cracking nuts[45]. However, footage selected for analysis was randomly sampled, representing a period of many years, in an attempt to amplify the signal-to-noise ratio. As such, the dataset produced here represents a critical longitudinal insight into the technological behaviour of a community presently on the cusp of extinction.

Third, the Bossou population structure hindered the collection of data from multiple individuals for the 20–35-year-old age range. As such, data were only obtained from one chimpanzee (Foaf), despite this being a potentially key period of stability in technological efficiency. This sample size of one meant that we could not compare the performance of individuals in this age bracket. More generally, across all ages,

there were very few individuals who overlapped in time, meaning that there were insufficient data for an inferential analysis of the stability of relative nut-cracking efficiency over time. Future research from field sites with larger community sizes is needed to further address the question of long-term stability of individual differences in technological efficiency.

## Conclusion

This study systematically assessed individual differences in nut-cracking in the Bossou chimpanzees using a long-term video archive of 25 years to longitudinally investigate these differences. Our results suggest reliable individual-level differences across four measures of nut-cracking efficiency, shedding light on the underlying cognitive and behavioural diversity in the Bossou chimpanzees. This research contributes to a growing body of evidence finding stable and reliable cognitive abilities in great apes, and points to potential variation in the development of this extractive foraging skill. Future research should seek to establish the factors driving this individual variation and its development over time.

## Methods

Ethical approval and permissions to conduct scientific research in the Bossou community were obtained by each contributor to the video archive from the Direction Générale de la Recherche Scientifique et de l'Innovation Technologique (DGERSIT) and the Institut de Recherche Environnementale de Bossou (IREB) in Guinea.

### Study site

Bossou is a village in south-eastern Guinea (7° 38′ 71.7″ N, 8° 29′ 38.9″ W), with a tropical wet seasonal climate and a predominant population of the Manon ethnic group[46]. The neighbouring chimpanzee community resides in primary and secondary forest, with a home range of 15 km², although their core area is 7 km². The chimpanzees in Bossou have been studied continuously since 1976, with 21 individuals being present[47]. Since then, the community has been declining in size[48]. The Bossou chimpanzees use a stone hammer-and-anvil composite to extract oil palm nuts[11,37,49], with individuals requiring complementary coordinated action of both hands to manoeuvre three objects (hammer, anvil and nut) during a nut-cracking bout[50]. The chimpanzees also crack experimentally introduced coula nuts[29].

### Study materials

The Bossou chimpanzees have been recorded in every dry season (December–February) since 1988, resulting in a long-term video archive of their behaviour. Researchers observed and videoed the chimpanzees in the 'outdoor laboratory'[29,40]—a 7 m × 20 m clearing in the core of the community's home range on Mount Gban (7° 38′ 41.5″ N, 8° 29′ 50.0″ W), which is passed through daily[26,29,40]. The outdoor laboratory is experimental in nature, with many of the available raw materials (with established weights and dimensions) and nuts being provisioned by the researchers[37]. Observations of behaviour within the outdoor laboratory occur from behind a grass screen along one edge of the clearing. All observation sessions were recorded using at least two standardized camera angles[37] (wide- and standard-angle lenses), optimizing the viewing angles.

### Measures

Five distinct aspects of nut-cracking efficiency were measured: (1) the time it took for one nut to be cracked open (bout duration), (2) the number of times the nut was hit with the hammer (strikes per nut), (3) the proportion of bouts ending in the whole kernel being extracted, a broken kernel being extracted or no kernel being extracted (success rate), (4) the number of times the hammer strike resulted in the nut being hit off the anvil (displacement rate) and (5) the number of times the hammer stone was changed or repositioned when trying to crack one nut (tool switch rate). The archival nature of the data meant that hammer size could not be

controlled for; however, the Bossou chimpanzees select stone tools based on their properties and attribute functions to the stones based on those features (for example, hammer stones are wider and lighter than those used for anvils)[26]. As such, hammers fall into a limited range of standard sizes and so there is likely little variation in hammer size selection.

## Data collection

Behavioural analysis was conducted using Behavioural Observation Research Interactive Software (BORIS, v. 7.11.1)[51]. Of the 1,185 videos in the Bossou archive from 1992 to 2017, 966 videos contained visible nut-cracking bouts by at least one individual, amounting to 832 observation hours. Where possible, the videos from the standard-angle lens data were used for analysis. If the behaviour was obscured in this footage, the equivalent video with the wide-angle lens was uploaded to attempt to observe the behaviour. If the behaviour remained unobservable, the bout(s) were excluded. No footage was collected in 2001 or 2011, and so these are absent from the dataset.

Data were collected for all post-ELP ($n = 21$) chimpanzees present during the study period, with bouts being recorded for each year each focal individual was cracking nuts. Multiple bouts (up to 20) per individual per year were recorded to establish the degree of within-individual variation in efficiency, while also producing more independent data points, allowing between-individual variation to be assessed. No statistical methods were used to pre-determine the sample size, but our sample size is similar to those reported in previous publications (refs. [1],[8]). The full data collection protocol can be found on page 13 of Supplementary Information.

## Statistical analysis

Data were collected from a total of 4,188 nut-cracking bouts. All bouts where coula nuts were cracked ($n = 281$) were excluded so that only data from native oil palm nuts were analysed. All bouts where the bout outcome had not been recorded ($n = 31$) were excluded, leaving 3,882 complete oil-palm nut-cracking bouts for analysis. All bouts where an infant was clinging to the focal subject were removed from analyses ($n = 210$), as this was exclusive to certain female chimpanzees (Fana, Fanle, Fotaiu, Jire, Pili, Velu, Vuavua and Yo) and so could reduce the internal validity of the findings by altering the efficiency of these individuals. Removing these bouts ensured that all individuals were compared under equal circumstances. This left 3,672 bouts for analysis.

Only data for bouts in which a kernel was retrieved were included in the analyses for bout duration and strikes per nut ($n = 3,367$). Excluding the 'Failed' bouts here ensured that the amount of time it took to access the energetic reward of the enclosed kernel was analysed. The full dataset ($n = 3,672$) was analysed for success rate, displacement rate and tool switch rate.

All analyses were performed using R (v. 4.3.2)[52] and RStudio (v. 2023.09.1+494) for MacOS. The significance level ($\alpha$) was 0.05 for all analyses. Multilevel models were constructed to test for individual differences in the five measures of efficiency. Individual chimpanzees comprised the random factor, while age and sex were included as fixed effects. Simple models (without random effects) were constructed and compared against the multilevel models (Supplementary Tables 2–6); ANOVAs were used to determine the model with the best fit. The ANOVAs (two-tailed) assessed whether including a random intercept for individual chimpanzees significantly decreased the model prediction error. The model with the smallest prediction error (AIC and −2-log-likelihood values) was selected for each of the five components of efficiency. To assess stability of relative efficiency within age cohorts over time, random slopes for the effects of age were added to models, when possible (see below).

The linear multilevel model for bout duration was fitted and assessed using the lme4[53] and lmerTest[54] packages, respectively. Initially, a simple linear multilevel model (subject as a random intercept) was constructed, with age and sex as fixed effects. The bout duration data were strongly right skewed, so it was log transformed for use in the linear multilevel model[55]. Random slopes for the effect of age were not possible in this model, as the model failed to converge when including this random effect structure.

Given that the count data for strikes per nut did not contain zeros, zero-truncated Poisson (multilevel) models were fitted using the truncated_poisson family in the glmmTMB package[56], with age and sex as fixed effects. However, the check_dispersion function from the performance package[57] detected overdispersion in the multilevel model (subject as a random intercept). As such, zero-truncated negative binomial (multilevel) models were fitted with quadratic parameterization. Random slopes for the effects of age were added to the model to assess stability of relative efficiency over time.

As the data for success rate were ordinal, cumulative link (multilevel) models were fitted using the ordinal package[58]. The outcome factor was ordered to Failed, Smash, Successful, to account for the increasing degrees of efficiency with each increasing level of outcome[38]. Hessian matrices were calculated for all models to calculate model summaries. The number of quadrature points used in the adaptive Gauss–Hermite quadrature approximation was set to 7. Age was originally included as a fixed effect but was removed as the model did not converge. Sex remained as a fixed effect. Random slopes for the effect of age were not possible in this model, as the model had a singularity issue when including this random effect structure.

Negative binomial (multilevel) models were fitted for displacement rate and tool switch rate using the glmmTMB package. All models specified zero-inflation as being equal for all observations, given the large number of zeros in the data where an individual did not displace the nut ($n = 2,909$) or switch their tools ($n = 3,308$) in a bout. Random slopes for the effects of age were added to the displacement rate model to assess stability of relative efficiency over time.

The individuals' random intercepts were ranked for the log bout duration, strikes per nut, success rate, displacement rate and tool switch rate models, and a correlation matrix was constructed to determine whether they represent the same underlying construct. The two-way ICC used to assess the reliability of individual variation was selected according to recommended guidelines[59].

Model assumptions were checked and revealed no issues (see page 17 of Supplementary Information and Supplementary Fig. 7).

## Reporting summary

Further information on research design is available in the Nature Portfolio Reporting Summary linked to this article.

## Data availability

All data can be found in the following public repository: https://osf.io/qw9ua.

## Code availability

All analysis code needed to reproduce the results and figures reported in the paper and Supplementary Information can be found in the following public repository: https://github.com/arranjdavis/chimpanzee_nut_cracking_efficiency.

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

## Acknowledgements

We thank the Direction Générale de la Recherche Scientifique et de l'Innovation Technologique (DGERSIT) and the Institut de Recherche Environnementale de Bossou (IREB) in Guinea for research authorization. The original video archive was digitized, organized and systematized by D. Schofield. Special thanks to all the field researchers and assistants at Bossou, specifically H. D. Camara, C. V. Mami, G. Zogbila, B. Zogbila, J. Doré, G. Goumy, T. Camara, P. Goumy, P. Cherif, J. G. Doré and M. Doré. We thank all the researchers who contributed to the Bossou video archive since its inception, especially D. Biro, M. Hayashi and C. Hobaiter. We thank A. Mielke for help in cataloguing the Bossou video archive. We thank D. Biro, K. Almeida-Warren and K. Koops for feedback. We thank Z. Dai and V. Martignac for help with independent, hypothesis-blind coding for the inter-rater reliability analyses. This research was supported by the University of Oxford's Clarendon Fund Scholarship (SFF1920_CB2_SSD_1153789) and the Boise Trust Fund (University of Oxford, UK) to S.B., a Junior Research Fellowship (Wadham College, University of Oxford) to A.J.D., and by the Ministry of Education, Culture, Sports, Science and Technology (MEXT), Japan (grant numbers 07102010, 12002009, 16002001, 20002001, 24000001 and 16H06283), to T.M. The funders had no role in study design, data collection and analysis, decision to publish or preparation of the paper.

## Author contributions

S.B. conceived of the study, designed and coordinated the study, collected data, analysed and visualized the data, and wrote the paper. E.C. provided supervision, participated in the design of the study and commented on the paper. A.J.D. analysed and visualized the data and commented on the paper. T.M. collected the original dataset and commented on the paper. S.C. collected the original dataset, provided supervision, participated in the design of the study and commented on the paper.

## Competing interests

The authors declare no competing interests.

## Additional information

**Correspondence and requests for materials** should be addressed to S. Berdugo.

# Reporting Summary

Please do not complete any field with "not applicable" or n/a. Refer to the help text for what text to use if an item is not relevant to your study.
For final submission: please carefully check your responses for accuracy; you will not be able to make changes later.

## Statistics

For all statistical analyses, confirm that the following items are present in the figure legend, table legend, main text, or Methods section.

| n/a | Confirmed | |
|---|---|---|
| ☐ | ☒ | The exact sample size (*n*) for each experimental group/condition, given as a discrete number and unit of measurement |
| ☐ | ☒ | A statement on whether measurements were taken from distinct samples or whether the same sample was measured repeatedly |
| ☐ | ☒ | The statistical test(s) used AND whether they are one- or two-sided *Only common tests should be described solely by name; describe more complex techniques in the Methods section.* |
| ☐ | ☒ | A description of all covariates tested |
| ☐ | ☒ | A description of any assumptions or corrections, such as tests of normality and adjustment for multiple comparisons |
| ☐ | ☒ | A full description of the statistical parameters including central tendency (e.g. means) or other basic estimates (e.g. regression coefficient) AND variation (e.g. standard deviation) or associated estimates of uncertainty (e.g. confidence intervals) |
| ☐ | ☒ | For null hypothesis testing, the test statistic (e.g. *F*, *t*, *r*) with confidence intervals, effect sizes, degrees of freedom and *P* value noted *Give P values as exact values whenever suitable.* |
| ☒ | ☐ | For Bayesian analysis, information on the choice of priors and Markov chain Monte Carlo settings |
| ☐ | ☒ | For hierarchical and complex designs, identification of the appropriate level for tests and full reporting of outcomes |
| ☐ | ☒ | Estimates of effect sizes (e.g. Cohen's *d*, Pearson's *r*), indicating how they were calculated |

*Our web collection on statistics for biologists contains articles on many of the points above.*

## Software and code

Policy information about availability of computer code

| | |
|---|---|
| Data collection | Data were collected using Behavioural Observation Research Interactive Software (BORIS, v. 7.11.1). |
| Data analysis | Data were analysed using custom code run in R (v. 4.3.2) and RStudio (v. 2023.09.1+494) for MacOS. The code can be found in the following public repository: https://github.com/arranjdavis/chimpanzee_nut_cracking_efficiency. The following packages were used in data analysis: lme4 (v. 1.1.34), lmerTest (v. 3.1.3), glmmTMB (v. 1.1.8), performance (v. 0.12.3), ordinal (v. 2022.11.16), irr (v. 0.84.1), car (v. 3.1.2), sure (v. 0.2.0), and influence.ME (v. 0.9.9). |

For manuscripts utilizing custom algorithms or software that are central to the research but not yet described in published literature, software must be made available to editors and reviewers. We strongly encourage code deposition in a community repository (e.g. GitHub). See the Nature Portfolio guidelines for submitting code & software for further information.

## Data

Policy information about availability of data

All manuscripts must include a data availability statement. This statement should provide the following information, where applicable:
- Accession codes, unique identifiers, or web links for publicly available datasets
- A description of any restrictions on data availability
- For clinical datasets or third party data, please ensure that the statement adheres to our policy

All data can be found in the following public repository: https://osf.io/qw9ua/

## Research involving human participants, their data, or biological material

Policy information about studies with [human participants or human data](). See also policy information about [sex, gender (identity/presentation), and sexual orientation]() and [race, ethnicity and racism]().

| | |
|---|---|
| Reporting on sex and gender | N/A |
| Reporting on race, ethnicity, or other socially relevant groupings | N/A |
| Population characteristics | N/A |
| Recruitment | N/A |
| Ethics oversight | N/A |

Note that full information on the approval of the study protocol must also be provided in the manuscript.

# Field-specific reporting

Please select the one below that is the best fit for your research. If you are not sure, read the appropriate sections before making your selection.

☐ Life sciences   ☐ Behavioural & social sciences   ☒ Ecological, evolutionary & environmental sciences

For a reference copy of the document with all sections, see [nature.com/documents/nr-reporting-summary-flat.pdf](nature.com/documents/nr-reporting-summary-flat.pdf)

# Life sciences study design

All studies must disclose on these points even when the disclosure is negative.

| | |
|---|---|
| Sample size | |
| Data exclusions | |
| Replication | |
| Randomization | |
| Blinding | |

# Behavioural & social sciences study design

All studies must disclose on these points even when the disclosure is negative.

| | |
|---|---|
| Study description | |
| Research sample | |
| Sampling strategy | |
| Data collection | |
| Timing | |
| Data exclusions | |
| Non-participation | |
| Randomization | |

# Ecological, evolutionary & environmental sciences study design

All studies must disclose on these points even when the disclosure is negative.

| | |
|---|---|
| Study description | Quantitative data were obtained through behavioural observation from a long-term video archive of wild chimpanzees cracking nuts using stone tools. |
| Research sample | The wild chimpanzee (Pan troglodytes verus) community of Bossou, Guinea (West Africa). |
| Sampling strategy | All chimpanzees aged 6+ within the 1992–2017 footage were sampled for this study. This amounted to 21 individuals. |
| Data collection | Data were collected from each year each individual was present in the archive. All nut-cracking data from the rare individuals (present in <25% of footage for a year) were collected, and data from other nut-cracking individuals present in that footage were collected. Videos were then randomly sampled from each year until up to 20 bouts were sampled for each individual for each year. |
| Timing and spatial scale | The research used the 1992–2017 footage from the Bossou video archive, which includes all years except 2001 and 2011. Footage was taken in the dry season (December–February) for each year. |
| Data exclusions | Nut-cracking bouts with infants clinging to the focal chimpanzee (n = 210) were excluded as this was exclusive to certain females. For two of the five models (bout duration, strikes per nut), 'failed' bouts (n = 305) were removed so that models assessed the time taken to extract the resource. Bouts of nut-cracking that involved coula nuts (Coula edulis, n = 281) were excluded so that only nut-cracking bouts involving native oil palm nuts (Elaeis guineensis) were analysed. All bouts where the bout outcome (Failed, Smash, Successful) was not recorded (n = 31) were excluded so that only bouts with complete information were analysed. |
| Reproducibility | Two hypothesis-blind, independent observers coded a subset of the footage to ensure the reliability of the data collected. Results indicated substantial-excellent agreement between coders. |
| Randomization | The study was a purely observational individual differences study and so no experimental manipulation was involved. As such, randomisation was not necessary. |
| Blinding | As there was no experimental manipulation, blinding was not necessary for this study. |

Did the study involve field work?  ☐ Yes  ☒ No

# Field work, collection and transport

| | |
|---|---|
| Field conditions | |
| Location | |
| Access & import/export | |
| Disturbance | |

# Reporting for specific materials, systems and methods

We require information from authors about some types of materials, experimental systems and methods used in many studies. Here, indicate whether each material, system or method listed is relevant to your study. If you are not sure if a list item applies to your research, read the appropriate section before selecting a response.

## Materials & experimental systems

| n/a | Involved in the study |
|---|---|
| ☒ | ☐ Antibodies |
| ☒ | ☐ Eukaryotic cell lines |
| ☒ | ☐ Palaeontology and archaeology |
| ☐ | ☒ Animals and other organisms |
| ☒ | ☐ Clinical data |
| ☒ | ☐ Dual use research of concern |
| ☒ | ☐ Plants |

## Methods

| n/a | Involved in the study |
|---|---|
| ☒ | ☐ ChIP-seq |
| ☒ | ☐ Flow cytometry |
| ☒ | ☐ MRI-based neuroimaging |

# Antibodies

| | |
|---|---|
| Antibodies used | |
| Validation | |

# Eukaryotic cell lines

Policy information about cell lines and Sex and Gender in Research

| | |
|---|---|
| Cell line source(s) | |
| Authentication | |
| Mycoplasma contamination | |
| Commonly misidentified lines (See ICLAC register) | |

# Palaeontology and Archaeology

| | |
|---|---|
| Specimen provenance | |
| Specimen deposition | |
| Dating methods | |
| ☐ Tick this box to confirm that the raw and calibrated dates are available in the paper or in Supplementary Information. | |
| Ethics oversight | |

Note that full information on the approval of the study protocol must also be provided in the manuscript.

# Animals and other research organisms

Policy information about studies involving animals; ARRIVE guidelines recommended for reporting animal research, and Sex and Gender in Research

| | |
|---|---|
| Laboratory animals | N/A |
| Wild animals | Wild chimpanzees (Pan troglodytes verus), but all from previously obtained video footage. |
| Reporting on sex | The sex of the chimpanzees is included in the results and Supplementary Information. Sex was included as a predictor variable in the models. |
| Field-collected samples | No field-collected samples were involved in this research. |
| Ethics oversight | Contributors to the Bossou video archive received research authorisation from the Direction Générale de la Recherche Scientifique et de l'Innovation Technologique (DGERSIT) and the Institut de Recherche Environnementale de Bossou (IREB) in Guinea. |

Note that full information on the approval of the study protocol must also be provided in the manuscript.

# Clinical data

Policy information about clinical studies
All manuscripts should comply with the ICMJE guidelines for publication of clinical research and a completed CONSORT checklist must be included with all submissions.

| | |
|---|---|
| Clinical trial registration | |
| Study protocol | |
| Data collection | |
| Outcomes | |

# Dual use research of concern

Policy information about dual use research of concern

## Hazards

Could the accidental, deliberate or reckless misuse of agents or technologies generated in the work, or the application of information presented in the manuscript, pose a threat to:

No | Yes
- [ ] | [ ] Public health
- [ ] | [ ] National security
- [ ] | [ ] Crops and/or livestock
- [ ] | [ ] Ecosystems
- [ ] | [ ] Any other significant area

## Experiments of concern

Does the work involve any of these experiments of concern:

No | Yes
- [ ] | [ ] Demonstrate how to render a vaccine ineffective
- [ ] | [ ] Confer resistance to therapeutically useful antibiotics or antiviral agents
- [ ] | [ ] Enhance the virulence of a pathogen or render a nonpathogen virulent
- [ ] | [ ] Increase transmissibility of a pathogen
- [ ] | [ ] Alter the host range of a pathogen
- [ ] | [ ] Enable evasion of diagnostic/detection modalities
- [ ] | [ ] Enable the weaponization of a biological agent or toxin
- [ ] | [ ] Any other potentially harmful combination of experiments and agents

# Plants

Seed stocks

Novel plant genotypes

Authentication

# ChIP-seq

## Data deposition

- [ ] Confirm that both raw and final processed data have been deposited in a public database such as GEO.

- [ ] Confirm that you have deposited or provided access to graph files (e.g. BED files) for the called peaks.

Data access links
*May remain private before publication.*

Files in database submission

Genome browser session
(e.g. UCSC)

## Methodology

Replicates

Sequencing depth

Antibodies

Peak calling parameters

Data quality

Software

# Flow Cytometry

## Plots

Confirm that:

☐ The axis labels state the marker and fluorochrome used (e.g. CD4-FITC).

☐ The axis scales are clearly visible. Include numbers along axes only for bottom left plot of group (a 'group' is an analysis of identical markers).

☐ All plots are contour plots with outliers or pseudocolor plots.

☐ A numerical value for number of cells or percentage (with statistics) is provided.

## Methodology

Sample preparation

Instrument

Software

Cell population abundance

Gating strategy

☐ Tick this box to confirm that a figure exemplifying the gating strategy is provided in the Supplementary Information.

# Magnetic resonance imaging

## Experimental design

Design type

Design specifications

Behavioral performance measures

Imaging type(s)

Field strength

Sequence & imaging parameters

Area of acquisition

Diffusion MRI          ☐ Used          ☐ Not used

## Preprocessing

Preprocessing software

Normalization

Normalization template

Noise and artifact removal

Volume censoring

## Statistical modeling & inference

Model type and settings

Effect(s) tested

Specify type of analysis: ☐ Whole brain ☐ ROI-based ☐ Both

Statistic type for inference

(See Eklund et al. 2016)

Correction

## Models & analysis

| n/a | Involved in the study |
|---|---|
| ☐ | ☐ Functional and/or effective connectivity |
| ☐ | ☐ Graph analysis |
| ☐ | ☐ Multivariate modeling or predictive analysis |

Functional and/or effective connectivity

Graph analysis

Multivariate modeling and predictive analysis

