## [Peer Review File · Nature Human Behaviour]

Reliable long-term individual variation in wild chimpanzee technological efficiency

Corresponding Author: Dr Sophie Berdugo

Version 0:

Decision Letter:

6th March 2024

Dear Ms Berdugo,

Thank you once again for your manuscript, entitled "Stable long-term individual variation in chimpanzee technological efficiency", and for your patience during the peer review process.

Your Article has now been evaluated by 3 referees. You will see from their comments copied below that, although they find your work of potential interest, they have raised quite substantial concerns. In light of these comments, we cannot accept the manuscript for publication, but would be interested in considering a revised version if you are willing and able to fully address reviewer and editorial concerns.

We hope you will find the referees' comments useful as you decide how to proceed. If you wish to submit a substantially revised manuscript, please bear in mind that we will be reluctant to approach the referees again in the absence of major revisions. We are committed to providing a fair and constructive peer-review process. Do not hesitate to contact us if there are specific requests from the reviewers that you believe are technically impossible or unlikely to yield a meaningful outcome.

To guide the scope of the revisions, the editors discuss the referee reports in detail within the team, including with the chief editor, with a view to (1) identifying key priorities that should be addressed in revision and (2) overruling referee requests that are deemed beyond the scope of the current study. We hope that you will find the prioritised set of referee points to be useful when revising your study. Please do not hesitate to get in touch if you would like to discuss these issues further.

1) Reviewers 1 and 2 both request that you examine stability/diversity across time. Editorially we agree that this is crucial and ask that you address this with new analyses.

2) Reviewer 3 notes that you could examine the relationship between individual efficiency in tool use and longevity/reproductive success using your data, and this analysis is much-needed in the field. Editorially we agree that this would significantly strengthen the paper. Please address the reviewer's concern by running and reporting these additional analyses (or by explaining why this is not possible with your data).

If you wish to submit a suitably revised manuscript, we would hope to receive it within 4 months. I would be grateful if you could contact us as soon as possible if you foresee difficulties with meeting this target resubmission date.

- Include a "Response to the editors and reviewers" document detailing, point-by-point, how you addressed each editor and referee comment. If no action was taken to address a point, you must provide a compelling argument. When formatting this document, please respond to each reviewer comment individually, including the full text of the reviewer comment verbatim followed by your response to the individual point. This response will be used by the editors to evaluate your revision and sent back to the reviewers along with the revised manuscript.
- Highlight all changes made to your manuscript or provide us with a version that tracks changes.

Link Redacted

Thank you for the opportunity to review your work. Please do not hesitate to contact me if you have any questions or would like to discuss the required revisions further.

Sincerely,

[REDACTED]

Reviewer expertise:

Reviewer #1: wild chimp behaviour, social organization, chimpanzee culture

Reviewer #2: wild chimp behaviour, chimpanzee tool use, cultural evolution

Reviewer #3: primate tool use, wild primates

REVIEWER COMMENTS:

Reviewer #1:

Remarks to the Author:

Berdugo and colleagues have coded and analyzed an impressive long-term data set of videos from the outdoor nut cracking lab at the Bossou chimpanzee field site in Guinea. The data set comprises a total of 3882 nut cracking bouts of 21 chimpanzees over 25 years. The authors extracted 4 efficiency measures from the coded video data set that they tested for individual variability and stability across measures. They find individual variability and they find that the efficiency measures are rank correlated. They conclude that their analysis shows longitudinal stability of these efficiency measures.

I see several problems with the study:

(1) The authors define 4 efficiency measures: bout duration, strikes per nut, displacement rate and tool switch rate. Bout duration and strikes per nut are highly correlated and therefore most likely not really independent measures (the more often you strike the nut the longer it takes). Displacement rate is not only dependent on the how forceful the nut is hit, but also if the nut shell is hollow / rotten or not. So with some correction one can potentially measure how well balanced the chimpanzee does place its force with the hammer. Tool switch can be interpreted in either way: either one is switching because there is a better tool available or became available or one is not switching because one has chosen the best tool already. So to be honest I do not see any expansion from what other studies have done.

(2) The data set is of 21 individuals with many repeated measures. To me this requires statistics allowing for repeated measures (e.g. mixed models). I was not really able to follow the statistical approach they were using, but to my understanding a mixed model would have been the right way to go here not the ANOVA (?).

(3) Despite the fact they do not control for ID while having repeated measures, they use the fixed effect p-values to argue for effects or no-effects. To my understanding the model structure does not allow for this.

(4) From the beginning the authors try to appear as if they look at longitudinal stability effects. Line 35 – 37: “to investigate whether individual differences in nut-cracking efficiency exist across the life span of chimpanzees aged ≥ 6 years.” To my understanding they do not do this. As shown in Figure 3 they rank the intercepts and correlate them across measures. They find that the measures are correlated with each other. However, I cannot see any analysis that would look at the stability of the measures across time. This could be done when looking for example at the random slopes within individuals across the years. However, I am unable to detect any similar analysis in the paper.

Now, I have to admit that the statistical procedure is mostly unclear to me. Therefore, I suggest to allow a statistician to judge the analysis and if that warrants the conclusions.

Here are some more specific comments:

Line 39 – 40: You may have not found sex differences in your study in Bossou, however that does not challenge any result from other field sites. Behavioural differences are anywhere across chimpanzee populations.

Line 100 – 105: the most convincing variation of tool use between populations is to me the example of honey dipping between Budongo and Kibale. Here there is a controlled field experiment showing the variability.

Line 115 – 116 same line 128: learning of nut cracking till 5 years of age? You should specify here that this is for Bossou. There are other numbers out for other study sites.

Line 136: how is a nut cracking bout defined?

Line 137: does this mean 9 males? Ages ?

Line 144 – 148: without a random factor ID for repeated measures the fixed factors do not really provide any helpful information.

Line 158 – 160: This means it is not individual consistency across time, but only across measures.

Line 170 “long-term stable” - we don't know, to my understanding not tested!

Line 242 – 245: arguing with ecological validity within a provisioned lab, although out door, I would be very careful.

Reviewer #2:

Remarks to the Author:

This paper describes stable and reliable individual differences in chimpanzee technological efficiency, specifically in nut-cracking tasks, across their lifespan. The authors argue that this variation directly impacts energy budgets, and potentially has significant

consequences for individual fitness and life history. The study emphasizes the importance of longitudinal data from long-term field sites for understanding cognitive and behavioral diversity across individual lifespans and between populations. Additionally, the research highlights the implications of individual variation in chimpanzee stone tool use for interpreting archaeological records of hominins. The research emphasizes the critical role of longitudinal data from extended field studies in elucidating primate behavior. Such data provide a unique opportunity to track the development and persistence of individual chimpanzees' tool-use strategies over an extended time. This offers generally valuable insights into the evolution of cognitive abilities among our closest relatives. The long-term data available from Bossou over 40 years is a unique opportunity to ask these research questions. Very few field sites are able to do so. Current research trends however are shifting in the opposite direction, where observations from the field often come from temporary sites that do not habituate the animals to the observer to minimize the risk of poaching and disease transmission. I would suggest including this in the discussion and adding to the analysis a part that shows how the stability of individual behavior over time (especially in the absence of age in the model, see below). The authors could contribute this way to an ongoing debate of what can be learned from "snapshots" of animal behavior and whether or not this is representative of a lifetime.

It would improve the paper if the authors could link their observations of tool use and their hypothesis of fitness gains to actual data from the field (number of offspring, age of disappearance). Literature on differences in tool use has not been able to be linked to fitness gains. The authors have a unique opportunity to do so.

Primates in line 33 need to be specified as non-human or defined above in line 28.

The statistical analysis is robust. To assess the validity of the author's findings the diversity of individuals over time needs to be displayed in the results section, especially as age had to be excluded from the model. Further, tool selection needs to be controlled for hammer size as well.

I would urge the authors to be more careful with the language of some of their statements. For example, in the abstract the paper starts with a generalization that needs to be rephrased as it is too specific (for example honey, termites, nuts...) are not proven to have been exploited, let alone initiated fundamental shifts in our evolution.

I don't understand why the authors repeat multiple times that this is the first study to show individual diversity. Generally, academic papers are the first to show something. If you want to say this once I guess it's fine, but reading it throughout the manuscript is not standard.

The authors discuss a difference across sexes in technological behavior. Can they please discuss these differences also in light of the rather unusual situation in Bossou regarding group size and lack of neighbors as this can play a role in the general diversity across chimpanzee groups.

Reviewer #3:

Remarks to the Author:

This is a well-structured and well-written manuscript in which the authors present the first longitudinal study of the stability and reliability of individual differences in the stone-assisted nut-cracking behavior of wild chimpanzees. The authors' main argument is that investigating the cognitive, behavioral, and environmental underpinnings of the ontogenetic process sustaining culturally-maintained technology has key implications for evolutionary fitness and life history. This research aims to further our understanding of the evolution of tool use in hominins.

The topic was well introduced. The background literature was appropriately reviewed (mixing seminal/historical and recent/up-to-date key articles) and critically addressed. The objectives were clearly laid out and ultimately reached.

Overall, the layout of the manuscript is logical, its tone is light, and its style is clear; these qualities result in an easy and pleasant read. Findings from previous research were clearly summarized and put into perspective in light of this study. The authors did an excellent job at making some relatively complex analyses simple, which makes the manuscript accessible to a non-expert audience (while maintaining the interest of experts in this field!). I have no major issues with the outline/structure and conclusion of the manuscript. The tables and figures are informative and relevant. This is an interesting manuscript that also fits an information gap.

Most importantly, the authors' thought-provoking proposition and innovative approach, if further refined, have the potential to significantly enhance our perspective on the development, mechanisms, functions, and evolution of lithic technology. Overall, I firmly believe that the authors' core argument is a key step towards a much-needed unified theoretical framework for the proximate and ultimate causes of tool use.

Before I can recommend this manuscript for publication, I just have one question that is actually not central to the manuscript but could significantly enhance its value. If the authors are willing to address this question in the Discussion section – even though they may not be able, or willing, to answer it in the present study – then I am happy to recommend this manuscript for publication.

Question: Lines 62-66, you address the potential benefits of higher proficiency and efficiency in extractive-foraging techniques (including tool use) in terms of evolutionary fitness, and you cite Biro et al.'s (2013) paper on tool use as adaptation.

However, in Biro et al.'s words "Surprisingly, we are unaware of any other studies that have attempted to document, directly, the fitness consequences of tool-use behaviour. [...] Perhaps most surprisingly, even among the long-studied chimpanzee populations of Gombe and Mahale in Tanzania and Bossou in Guinea, which have records of tool use and genealogies extending back many decades, no published research has examined the link between tool-use frequency (or competence) and fitness. This, for now, remains a major research challenge in our field."

Considering your 25-year longitudinal data set, it may be possible to tackle this essential problem for the first time. Could you please explain to the readers why you didn't use your data set to conduct these much-needed analyses testing the effect of individual variation in tool-assisting nut-cracking efficiency on longevity and reproductive success?

Version 1:

Decision Letter:

Our ref: NATHUMBEHAV-23113916A

13th August 2024

Dear Dr. Berdugo,

Thank you for submitting your revised manuscript "Stable long-term individual variation in chimpanzee technological efficiency" (NATHUMBEHAV-23113916A). It has now been seen by the original referees and their comments are below. As you can see, the reviewers find that the paper has improved in revision. Reviewer 1 has some remaining concerns and suggestions for improvement. We will therefore be happy in principle to publish it in Nature Human Behaviour, pending minor revisions to satisfy Reviewer 1's final requests and to comply with our editorial and formatting guidelines.

We are now performing detailed checks on your paper and will send you a checklist detailing our editorial and formatting requirements within two weeks. Please do not upload the final materials and make any revisions until you receive this additional information from us.

Sincerely,

[REDACTED]

Reviewer #1 (Remarks to the Author):

Thank you for revising the paper and taking the concerns into account. I can see that several of the comments of my review have been addressed. The result of the paper is now that there is individual variation in nut-cracking efficiency and the individual variation is stable in a subset of 7 individuals over time. Why the authors look in their analysis (see Fig 4) only at random slopes of 7 individuals and not for 21 remains unclear.

Although I agree with the authors' general conclusion that due to individual variation we need to look at individual performance for energy intake and avoid group averages when looking at fitness related questions. However, the authors' analysis (also incomplete) shows in Fig 4 nicely that individual variation is much larger in younger years. In contrast once in the older ages (35 years and older) variation is actually not that strong anymore. This makes entirely sense for tasks that need many years to be learned and improved. We know from other sides that tool use in chimpanzees gets improved into adulthood and nut cracking efficiency has not plateaued yet even when being 10 years old. The crucial data between 20 and 35 are represented by one single individual. Here readers are unable to judge what is happening.

Age seems to have a strong effect on the efficiency, being the significant predictor in four models. Since random factors and slopes have been considered, this effect is not an effect of individual variation. The direction of the effect should be reported – and not ignored in results and abstract.

We also would expect to have actually an increase of efficiency in younger ages, but than also a decrease of efficiency in old age, something that has not been modeled (which would require a squared term for age).

I understand that the authors have used a linear mixed model approach. Why they have chosen a "linear (mixed) model" approach is not clear to me. Diverting from the usual nomenclature creates suspicions that are most likely misplaced. Therefore, I strongly suggest to change this for clarity.

Reviewer #2 (Remarks to the Author):

The authors incorporated my suggestions (or explained why this is not possible) and I believe the paper is now a good contribution to our field and ready for publication.

Lydia Luncz

Reviewer #3 (Remarks to the Author):

I am appreciative of the time and work that the authors have put into their detailed response to the reviewers' comments and extensive revision of the manuscript. In the process, I thank them for addressing my question to my satisfaction.

I am now happy to state that the manuscript has improved accordingly. It is a pleasure to read. I can therefore wholeheartedly recommend it for publication in Nature – Human Behaviour.

Version 2:

Decision Letter:

Dear Dr Berdugo,

We are pleased to inform you that your Article "Reliable long-term individual variation in wild chimpanzee technological efficiency", has now been accepted for publication in *Nature Human Behaviour*.

Please note that *Nature Human Behaviour* is a Transformative Journal (TJ). Authors may publish their research with us through the traditional subscription access route or make their paper immediately open access through payment of an article-processing charge (APC). Authors will not be required to make a final decision about access to their article until it has been accepted. [Find out more about Transformative Journals](https://www.springernature.com/gp/open-research/transformative-journals)

We welcome the submission of potential cover material (including a short caption of around 40 words) related to your manuscript; suggestions should be sent to *Nature Human Behaviour* as electronic files (the image should be 300 dpi at 210 x 297 mm in either TIFF or JPEG format). Please note that such pictures should be selected more for their aesthetic appeal than for their scientific content, and that colour images work better than black and white or grayscale images. Please do not try to design a cover with the *Nature Human Behaviour* logo etc., and please do not submit composites of images related to your work. I am sure you will understand that we cannot make any promise as to whether any of your suggestions might be selected for the cover of the journal.

With best regards,

[REDACTED]

P.S. Click on the following link if you would like to recommend Nature Human Behaviour to your librarian
<http://www.nature.com/subscriptions/recommend.html#forms>

** Visit the Springer Nature Editorial and Publishing website at http://editorial-jobs.springernature.com?utm_source=ejp_NHumB_email&utm_medium=ejp_NHumB_email&utm_campaign=ejp_NHumB for more information about our career opportunities. If you have any questions please click [here](mailto:editorial.publishing.jobs@springernature.com). **

Reviewer #1:

(1) The authors define 4 efficiency measures: bout duration, strikes per nut, displacement rate and tool switch rate. Bout duration and strikes per nut are highly correlated and therefore most likely not really independent measures (the more often you strike the nut the longer it takes). Displacement rate is not only dependent on the how forceful the nut is hit, but also if the nut shell is hollow / rotten or not. So with some correction one can potentially measure how well balanced the chimpanzee does place it force with the hammer. Tool switch can be interpreted in either way: either one is switching because there is a better tool available or became available or one is not switching because one has chosen the best tool already. So to be honest I do not see any expansion from what other studies have done.

We used five measures of nut-cracking efficiency for our research, all supported by previous research (see Table 1). One of the ways our research expands on these other studies is by testing the reliability of these measures. We found that four of the five measures were internally consistent, meaning that if an individual ranked highly on one of the measures, they also ranked highly on the other three. This was further supported by the good absolute agreement between these four measures shown by an intra-class correlation.

We have now included the following into our discussion to clarify why these measures can be considered as independent (lines 215–222): **‘The ranked random intercepts for *bout duration*, *strikes per nut*, *success rate*, and *displacement rate* were closely correlated, but the raw scores were not perfectly correlated. For example, the correlation between the ranked random intercepts for *bout duration* and *strikes per nut* was greater than the correlation between their raw scores ($r = 0.99$ and $r = 0.756$, respectively). Indeed, longer bouts do not necessarily equate to more strikes per nut, with bout duration also being extended due to factors such as greater distractibility or taking longer pauses between strikes. This supports the view that each measure captures distinct, but internally consistent, aspects of what can be termed “efficiency”.’**

(2) The data set is of 21 individuals with many repeated measures. To me this requires statistics allowing for repeated measures (e.g. mixed models). I was not really able to follow the statistical approach they were using, but to my understanding a mixed model would have been the right way to go here, not the ANOVA (?).

We have taken a mixed model approach to all regressions, with individual as a random intercept term to account for the repeated measures nature of our design. Our use of ANOVAs was to determine whether including the random intercept term decreased the prediction error (AIC and -2-log-likelihood values) for each model (it did).

(3) Despite the fact they do not control for ID while having repeated measures, they use the fixed effect *p*-values to argue for effects or no-effects. To my understanding the model structure does not allow for this.

As explained in our response to the previous comment, we have included the subject (individual chimpanzee) as a random intercept term. As such, we believe we are able to use the *p*-values to determine whether there are significant fixed effects of age and sex.

(4) From the beginning the authors try to appear as if they look at longitudinal stability effects. Line 35 – 37: “to investigate whether individual differences in nut-cracking efficiency exist across the life span of chimpanzees aged ≥ 6 years.” To my understanding they do not do this. As shown in Figure 3 they rank the intercepts and correlate them across measures. They find that the measures are correlated with each other. However, I cannot see any analysis that would look at the stability of the measures across time. This could be done when looking for example at the random slopes within individuals across the years. However, I am unable to detect any similar analysis in the paper.

Thank you for this suggestion. We have now incorporated this into the manuscript, having performed a new analysis where we include random slopes for age for each individual. Given that age had a significant effect in our first analyses, we did not predict stability in an individual’s performance for each efficiency measure across the years. Instead, we sought to determine whether individuals’ relative efficiency (compared to others of a similar age) persisted over time. For this, we looked at the four reliable measures of efficiency (*bout duration*, *strikes per nut*, *success rate*, and *displacement rate*) across seven adult chimpanzees aged 11–40. We chose this subset of adults as previous research, along with the present findings, established that young chimpanzees continue to become more efficient at the skill until around age 10 (Matsuzawa, 1994). Moreover, new research has found inter-individual variation in rates of declining nut-cracking efficiency in old age (40+) in Bossou (Howard-Spink E., Matsuzawa T., Carvalho S., Hobaiter C., Almeida-Warren K., et al., unpublished manuscript).

We have now added the following text to the results section of the manuscript (lines 189–193): ‘**We were able to create models with random slopes for age for *strikes per nut* and *displacement rate* (models for *bout duration* and *success rate* did not converge or had issues with singularity, respectively). Results are shown in Figure 4. Although more data is needed for inferential statistics, we note that estimated random slopes for individuals within age cohorts do not intersect, suggesting little change in relative efficiency over time within these groups.**’

We also discuss these findings in lines 250–260: ‘**Moreover, our results suggest that relative inter-individual nut-cracking efficiency (in terms of number of strikes per nut and the displacement rate) may be stable over time during the APP. Although data are relatively sparse—our models included only seven individuals—we were able to estimate the trajectory of individuals’ nut-cracking efficiency over time within age cohorts who overlapped in the years they were present in the Bossou archive. Model estimates suggest that an individual’s efficiency relative to others in its cohort**

persisted across overlapping ages; individuals' predicted random slopes for the effects of age on the efficiency outcomes did not intersect. These model predictions are thus consistent with the assertion that individuals' relative efficiency ranks remain stable over time. However, given the relatively small sample of individuals, these findings should be viewed as suggestive, and an avenue for future research.'

Here are some more specific comments:

Line 39 – 40: You may have not found sex differences in your study in Bossou, however that does not challenge any result from other field sites. Behavioural differences are anywhere across chimpanzee populations.

We have now removed this statement.

Line 100 – 105: the most convincing variation of tool use between populations is to me the example of honey dipping between Budongo and Kibale. Here there is a controlled field experiment showing the variability.

Thank you for highlighting this study. We have now included it in lines 99–101: **'Chimpanzees in Kibale National Park, Uganda, used sticks to extract experimentally introduced honey, whereas chimpanzees in Budongo National Park, Uganda, either used their fingers or leaf sponges'.**

Line 115 – 116 same line 128: learning of nut cracking till 5 years of age? You should specify here that this is for Bossou. There are other numbers out for other study sites.

We have now specified that this is for Bossou.

Line 136: how is a nut cracking bout defined?

We have now added in the definition of a nut-cracking bout to line 136–138: **'Bouts are the continuous periods (in seconds) of nut-cracking whereby the individual strikes a single nut on an anvil with a hammer stone involving the same hand, bodily posture, hand grip, and nut'.**

Line 137: does this mean 9 males? Ages ?

Thank you for noticing this omission. We have now edited this to **'21 chimpanzees (12 females and 9 males; ages 6–60)'**.

Line 144 – 148: without a random factor ID for repeated measures the fixed factors do not really provide any helpful information.

As explained in our response to the second point above, we have included the subject (individual chimpanzee) and a random intercept term.

Line 158 – 160: This means it is not individual consistency across time, but only across measures.

Our measure of reliability assesses the internal consistency of the different efficiency measures to establish whether they are reliable metrics for determining an individual's stone tool use efficiency. Our measure of stability assesses the individual differences in efficiency across time. We have clarified our definitions of reliability and stability in lines 131–133: **'We also assess the *reliability* (internal consistency across efficiency measures) and *stability* (whether individual differences hold over time) of such differences in a wild primate population.'**

Line 170 “long-term stable” - we don't know, to my understanding not tested!

As explained above, this has now been assessed via inclusion of the random slope plots for *strikes per nut* and *displacement rate* (Figure 4).

Line 242 – 245: arguing with ecological validity within a provisioned lab, although outdoor, I would be very careful.

We have now included that the nuts and stones were locally-sourced, to support the claim of ecological validity.

Reviewer #2:

The long-term data available from Bossou over 40 years is a unique opportunity to ask these research questions. Very few field sites are able to do so. Current research trends however are shifting in the opposite direction, where observations from the field often come from temporary sites that do not habituate the animals to the observer to minimize the risk of poaching and disease transmission. I would suggest including this in the discussion and adding to the analysis a part that shows how the stability of individual behavior over time (especially in the absence of age in the model, see below). The authors could contribute this way to an ongoing debate of what can be learned from “snapshots” of animal behavior and whether or not this is representative of a lifetime.

Thank you very much for the suggestion to include a discussion of the importance of long-term data on animal behaviour, and what can be learned from such data compared to short-term “snapshots”. We have now added this point to lines 289–291: **“This reiterates the importance of conducting long-term investigations of long-lived primates as data from short-term studies may not be representative of behaviour over the lifespan.”**

Also, thank you for your recommendation to analyse the stability of individual behaviour over time. We have now incorporated this into the manuscript (explained in detail in our response to the same suggestion by Reviewer #1, above), and we agree that this improves our contribution. Lastly, to clarify, age is included as a fixed effect in four of the five models (it is only absent from the success rate model given model convergence issues).

It would improve the paper if the authors could link their observations of tool use and their hypothesis of fitness gains to actual data from the field (number of offspring, age of disappearance). Literature on differences in tool use has not been able to be linked to fitness gains. The authors have a unique opportunity to do so.

Thank you for your suggestion. We agree that this is an important question to be addressed. However, it is not possible to establish a causal link between nut-cracking efficiency and longevity based on the data available. If less efficient nutcrackers died earlier, it would not be possible to distinguish whether this is a consequence of relatively poor technological efficiency, or, for example, their low efficiency being an indicator of another factor affecting survival (e.g., general weakness or ill-health). We have now added this explanation into our discussion, as also recommended by Reviewer 3, which we hope addresses this concern.

Primates in line 33 need to be specified as non-human or defined above in line 28.

We have now added this specification (now in line 30).

The statistical analysis is robust. To assess the validity of the author's findings the diversity of individuals over time needs to be displayed in the results section, especially as age had to be excluded from the model. Thank you, we are happy that you find our statistical analyses robust. As above, we would like to clarify that age is included as a fixed effect in the models for four of the five efficiency measures.

Further, tool selection needs to be controlled for hammer size as well.

Thank you for this suggestion. Previous research has established that the Bossou chimpanzees select hammer stone tools based on their properties (length, width, height, and weight) and attribute functions to the stones based on those features (e.g., if larger it becomes an anvil and if smaller it will be discarded or, very rarely, used as a wedge; Carvalho et al., 2008). The stones used in the hammer-and-anvil composites are also preferentially selected and often reused (Carvalho et al., 2009). Given these previous findings, in the case of Bossou, hammer stones fall into a very limited range of standard sizes and there is little variation in hammer size selection. We have now included this point into the manuscript (lines 371–375): **‘The archival nature of the data meant that hammer size could not be controlled for; however, the Bossou chimpanzees select stone tools based on their properties and attribute functions to the stones based on those features (e.g., hammer stones are wider and lighter than those used for anvils)²⁶. As such, hammers fall into a limited range of standard sizes and so there is likely little variation in hammer size selection.’**

I would urge the authors to be more careful with the language of some of their statements. For example, in the abstract the paper starts with a generalization that needs to be rephrased as it is too specific (for

example honey, termites, nuts...) are not proven to have been exploited, let alone initiated fundamental shifts in our evolution.

Thank you very much for this comment. We have now removed this from the abstract and have also checked the language throughout the manuscript, tempering our statements where they had been too strong.

I don't understand why the authors repeat multiple times that this is the first study to show individual diversity. Generally, academic papers are the first to show something. If you want to say this once I guess it's fine, but reading it throughout the manuscript is not standard.

Thank you, we have now removed all comments stating this is the first study to show various findings.

The authors discuss a difference across sexes in technological behavior. Can they please discuss these differences also in light of the rather unusual situation in Bossou regarding group size and lack of neighbors as this can play a role in the general diversity across chimpanzee groups.

Thank you for raising this important point. We have now referred to this in our discussion (lines 283–286), which now reads ‘This finding contradicts previous research establishing a female bias in technological behaviour across the genus *Pan*³⁹. **This may reflect circumstances specific to the Bossou chimpanzees. Indeed, Bossou is an isolated community that has been decreasing in size, with both factors potentially contributing to unusual patterns compared to other populations.**’

Reviewer #3:

Considering your 25-year longitudinal data set, it may be possible to tackle this essential problem for the first time. Could you please explain to the readers why you didn't use your data set to conduct these much-needed analyses testing the effect of individual variation in tool-assisting nut-cracking efficiency on longevity and reproductive success?

Thank you for highlighting this very important question. We have added the following paragraph to the discussion (lines 270–278), which explains why we have not tested this with our data set: ‘**This research did not seek to directly test whether increased nut-cracking efficiency corresponded to relative fitness gains as the presence of confounds hinders our ability to establish a causal link between tool use efficiency and longevity or reproductive success. For example, a shorter life span of a relatively less efficient individual could be the result of technological inefficiency or another factor that negatively impacts both tool use efficiency and longevity (e.g., physical weakness or ill-health). Research specifically investigating the fitness consequences of individual differences in technological efficiency should consider the broader energy budgets and expenditure, as relatively inefficient nutcrackers may make up for their energy disadvantages through increased efficiency in other tasks.**’

Reviewer #1:

Thank you for revising the paper and taking the concerns into account. I can see that several of the comments of my review have been addressed. The result of the paper is now that there is individual variation in nut-cracking efficiency and the individual variation is stable in a subset of 7 individuals over time. Why the authors look in their analysis (see Fig 4) only at random slopes of 7 individuals and not for 21 remains unclear.

Many thanks for taking the time to review our revised manuscript, and we are happy that you are satisfied that we have addressed your comments. Thank you for highlighting this lack of clarity in the results section. We have now included in the following clarification as to why we only added in the random slope of age for 7 of the 21 chimpanzees (lines 187–191): **“By looking only at individuals with at least three years of data, we were able to determine whether the random slope lines intersected one another or not, independent of the effects of age on efficiency reported above. In other words, we asked: even if individual X’s efficiency improved or worsened over time, were they always more or less efficient than individual Y?”**

Age seems to have a strong effect on the efficiency, being the significant predictor in four models. Since random factors and slopes have been considered, this effect is not an effect of individual variation. The direction of the effect should be reported – and not ignored in results and abstract.

Thank you for noticing that we had not reported the direction of the age effect in the results. We have now added this in lines 150–155: “Age had a significant **positive** fixed effect on *bout duration* ($t = 7.724, p < 0.001$) and *strikes per nut* ($z = 8.51, p < 0.001$), and a significant **negative** fixed effect on *displacement rate* ($z = -2.40, p = 0.0164$) and *tool switch rate* ($z = -2.462, p = 0.0138$).” We have also added the effect of age to the abstract.

We also would expect to have actually an increase of efficiency in younger ages, but than also a decrease of efficiency in old age, something that has not been modeled (which would require a squared term for age).

We have now included an explanation and a figure in the ‘Assumption checks’ section in our Supplementary Information that explains that the linearity assumption is met for the age term (lines 319–323): **“For the four models with age included as a fixed effect (bout duration, strikes per nut, displacement rate, tool switch rate), we checked the assumption that age was linearly related to the outcome measure by plotting the model residuals against the predictor variable. We found that age appears to be linearly related to our outcome variables (see Figure S7), and as such we did not need to square the age term in the models.”**

I understand that the authors have used a linear mixed model approach. Why they have chosen a “linear (mixed) model” approach is not clear to me. Diverting from the usual nomenclature creates suspicions that are most likely misplaced. Therefore, I strongly suggest to change this for clarity.

We have now gone through the reporting of our statistical modelling and ensured consistency in how the model approach has been reported. The statement of “linear (mixed) model” that Reviewer #1 has noted is only in the context of “generalised linear (mixed) models (GL(M)Ms)” to define what the second ‘M’ in GLMM is, given that we compared GLMs against GLMMs for the relevant outcome measures.

Reviewer #2:

The authors incorporated my suggestions (or explained why this is not possible) and I believe the paper is now a good contribution to our field and ready for publication.

Thank you, we are happy that you are satisfied with the revisions we made in response to your suggestions.

Reviewer #3:

I am appreciative of the time and work that the authors have put into their detailed response to the reviewers’ comments and extensive revision of the manuscript. In the process, I thank them for addressing my question to my satisfaction.

I am now happy to state that the manuscript has improved accordingly. It is a pleasure to read. I can therefore wholeheartedly recommend it for publication in Nature – Human Behaviour.

Thank you for recognising the work we put into revising the manuscript, and we appreciate your time in reviewing it.